# Diffusion$^2$: Turning 3D Environments into Radio Frequency Heatmaps

## Abstract

Modeling radio frequency (RF) signal propagation is essential for understanding the environment, as RF signals offer valuable insights beyond the capabilities of RGB cameras, which are limited by the visible-light spectrum, lens coverage, and occlusions. It is also useful for supporting wireless diagnosis, deployment, and optimization. However, accurately predicting RF signals in complex environments remains a challenge due to interactions with obstacles such as absorption and reflection. We introduce **Diffusion**$^2$, a diffusion-based approach that uses 3D point clouds to model the propagation of RF signals across a wide range of frequencies, from Wi-Fi to millimeter waves. To effectively capture RF-related features from 3D data, we present the *RF-3D Encoder*, which encapsulates the complexities of 3D geometry along with signal-specific details. These features undergo multi-scale embedding to simulate the actual RF signal dissemination process. Our evaluation, based on synthetic and real-world measurements, demonstrates that **Diffusion**$^2$ accurately estimates the behavior of RF signals in various frequency bands and environmental conditions, with an error margin of just 1.9 dB and 27x faster than existing methods, marking a significant advancement in the field. Refer to https://rfvision-project.github.io/ for more information.

## 1 Introduction

**Motivation:** Generative AI has reached remarkable milestones, as evidenced by ChatGPT (Achiam et al., 2023) and more recently Sora (Brooks et al., 2024). In particular, Sora has captivated the field with its ability to generate stunningly realistic videos that follow the laws of physics. We are driven by a fundamental question: *Can generative AI comprehend beyond the visible-light spectrum?*

In this paper, we specifically explore the use of generative AI to accurately estimate radio frequency (RF) heatmaps for 3D environments. An RF heatmap visualizes the distribution of signal strength across various locations within a given space, providing a comprehensive overview of wireless coverage and signal behavior. Our goal is to leverage generative models to predict these heatmaps with high fidelity, even under complex and dynamic environmental conditions.

The motivation behind this initiative stems from the diverse and critical applications that reliable RF heatmaps can enable across multiple domains. These include optimizing access point (AP) placement, advanced transmitter and receiver configuration, facilitating smart environments and IoT deployments, and automating site surveys and wireless diagnosis (Zheng et al., 2019; Chen & Zhang, 2023).

Although the propagation of RF signals in free space can be modeled using Maxwell's equations and the Friis transmission equation, real-world scenarios introduce numerous obstacles that disrupt the radiance field (Yun & Iskander, 2015). Environmental obstacles cause various effects, such as absorption, diffraction, reflection, and scattering. For example, scattering occurs when the RF signal interacts with objects or surface irregularities, resulting in the signal being redirected in multiple directions. Moreover, the topology and material properties of objects in the environment further complicate the understanding of signal propagation. Understanding and addressing these complexities is pivotal in our quest to generate accurate and reliable RF signal heatmaps for practical applications.

**Existing work:** Several studies have applied machine learning (ML) to estimate the RF signal at a receiver (Chi et al., 2024; Zhao et al., 2023b; Chen & Zhang, 2023). For instance, NeRF$^2$ (Zhao et al., 2023b) combines knowledge of the physical wave signal with NeRF (Mildenhall et al., 2021) to compute the strength of the wireless signal at a given location. Although recent work has shown promising results, both rely on pre-measured signal data in a specific environment (*e.g.*, 4k measurements). This incurs significant computational and pre-measurement costs, severely limiting their ability to generalize beyond experimental sites. Moreover, if there is a change in the location of

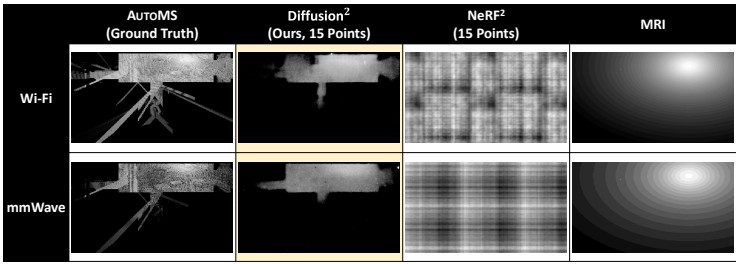

Figure 1: Results for **Diffusion**[2], AUTOMS (Ma et al., 2024), NeRF[2] (Zhao et al., 2023b), and MRI (Shin et al., 2014) for one example environment at two frequencies. The transmitter is located in the upper right corner of the room. 5.16 GHz and 77 GHz are used for Wi-Fi and millimeter wave (mmWave). **Diffusion**[2] and NeRF[2] are tested using the same 15 pre-measurements.

an object or a shift in the operating frequency, a large volume of new measurements must be collected to retrain the model.

Significant efforts have been made in environment modeling approaches that use 3D geometry, such as LiDAR point clouds or video footage, to simulate RF signal propagation based on physical laws (Wireless InSite, 2025; Ma et al., 2024). However, existing ray-tracing simulators struggle to balance high accuracy and efficiency. For example, Wireless InSite (Wireless InSite, 2025), a widely used commercial ray-tracing software, requires over 1.5 hours to estimate signals in a small room with 4,140 receivers. In general, the computational complexity of ray tracing-based approaches increases significantly with the number of receivers, making them less efficient for larger-scale environments.

**Our approach:** To address the challenges mentioned above, we introduce a novel approach, **Diffusion**[2], which transforms a 3D model of an environment into an RF heatmap. Specifically, it begins by capturing a 3D model of the environment using a smartphone application (*e.g.*, the Polycam (Polycam, 2025)), utilizing the LiDAR sensor available on mobile devices (*e.g.*, iPhones and iPads). This step takes approximately one minute. The 3D model and RF features are then fed into our neural network, which employs a diffusion model to generate the RF heatmap. The diffusion approach simplifies complex optimization problems into probabilistic calculations over multiple steps, thereby reducing the difficulty of learning (Ho et al., 2020; Song et al., 2020; Saharia et al., 2022).

The motivation for leveraging diffusion models for RF signal map generation lies in two key factors. First, diffusion models exhibit remarkable resilience to the inherent uncertainties of real-world environments caused by unobservable variables. Their multi-step probabilistic framework allows for generating results that closely adhere to the principle of maximum likelihood estimation, making them well-suited to handle complex environments. Second, despite being inherently stochastic, diffusion models offer excellent controllability. They can flexibly incorporate multi-dimensional and rich control parameters, enabling the generation of RF signals that accurately reflect the physical world, based on environmental details such as 3D geometry and RF data.

Specifically, a diffusion model is a generative ML framework designed to create new data samples. It operates in two phases: the forward diffusion process, where Gaussian noise is gradually added to the data until it is transformed into pure noise, and the reverse process, which reconstructs the original data from the noise. During training, the diffusion model learns the underlying distribution of the training data and refines its ability to effectively denoise, allowing it to generate realistic RF heatmaps that mirror the complexities of real-world signal propagation.

To apply the diffusion model for generating an RF heatmap corresponding to a 3D environment model, we leverage *conditioning* during the diffusion process. Conditioning is a technique that enables the generation of samples that meet specific criteria (Wang et al., 2024; Chen et al., 2023a; Dai et al., 2023). Each step of the diffusion process learns the conditional probability guided by the conditioning signal, ensuring that the generated output not only conforms to the RF signal distribution but also satisfies predefined conditions. To ensure that the generated RF signal map accurately reflects real-world outcomes, we propose the *RF-3D Encoder*, which extracts features from the 3D environment model and RF-related information. These features serve as conditions during the reverse diffusion process. By fine-tuning the model parameters, the generation process is optimized to produce samples that align with the desired criteria.

Beyond generating an RF heatmap from a static 3D model of the environment, it is also valuable to create a dynamic RF heatmap video as the 3D environment changes (*e.g.*, a human is moving). Video heatmap generation can greatly benefit various applications, including network provisioning, diagnosis, and wireless sensing, in dynamic environments. Inspired by Sora (Brooks et al., 2024)'s innovative video diffusion results, we extend our image diffusion to video diffusion, dynamically adapting to environmental changes, such as human locomotion, by incorporating temporal layers. These layers enable the interaction of our features and RF signal maps across multiple frames.

**Diffusion**[2] advances the state of the art by leveraging a 3D environment model with only a handful of pre-measurements, as illustrated in Fig. 1. It offers several distinct advantages over existing work: (1) It achieves high accuracy with minimal signal measurements (*e.g.*, 15 measurements in our evaluation) and eliminates the need for detailed information about the surrounding objects. In comparison, RF-Diffusion (Chi et al., 2024) and NeRF[2] (Zhao et al., 2023b) require thousands of measurements. (2) It enables fast computation (*e.g.*, processing over 200,000 receivers (RXs) in under one second), achieving a 27× speedup over AUTOMS (Ma et al., 2024) and a 33× speedup over NeRF[2]. (3) It can generate RF heatmaps across multiple frequencies, a capability that is highly valuable for operational tasks such as channel allocation and interference management. (4) It can transform both static 3D scenes into RF heatmap images and dynamic 3D scenes into RF heatmap videos. Our contributions are as follows:

- To the best of our knowledge, **Diffusion**[2] is the first generative diffusion model designed to estimate RF signal propagation using a 3D model of an environment. It is highly accurate, fast, easy to use, and generalizable, while supporting both Wi-Fi and mmWave frequencies.

- We propose the *RF-3D Encoder* for efficient feature extraction from 2D, 3D, and RF modalities, and the *RF-3D Pairing Block*, which enables effective cross-modal integration during diffusion.

- **Diffusion**[2] supports video diffusion, enabling fast adaptation to dynamic environmental shifting.

- We conduct extensive experiments across multiple frequencies with over 55k+ synthetic rooms and validate the robustness using real-world measurements. Our results show high accuracy within 1.9 dB while inferring over 27× faster than others, achieving real-time speed.

## 2 RELATED WORK

**Ray tracing.** MRI (Shin et al., 2014) estimates received signal strength indicator (RSSI) using a simple propagation model. Deng et al. (2017) survey hardware acceleration for ray tracing. Wireless InSite (Wireless InSite, 2025), a commercial software, and AUTOMS (Ma et al., 2024), a recent optimization using software and hardware, both utilize 3D point clouds for RF heatmaps.

Despite decades of great effort on ray tracing, it still faces several key challenges: 1) High computational demands and scalability issues persist, as the computation cost increases rapidly with the number of receivers. 2) It requires material information about each object, such as the reflection and attenuation coefficients, which are difficult to obtain in the real world. 3) Accurately modeling complex physical phenomena, such as soft diffraction, scattering due to edges, penetration through complex objects, and near-field propagation, remains an ongoing challenge.

**ML approaches.** To address the limitations of ray tracing, various ML approaches have been proposed. For instance, CGAN (Parralejo et al., 2021) uses a conditional generative adversarial network to directly predict RSSI values, eliminating the need for a specific physical model. NeRF[2] (Zhao et al., 2023b) introduces a deep learning framework designed to model wireless channels, integrating the physical model of electromagnetic wave transmission into the channel learning process. NeRF[2] supports various application-layer tasks, such as indoor localization and massive MIMO communication. However, NeRF[2] requires a large volume of measurements of the environment to train the model. If the environment changes, new data should be collected, and the model needs to be retrained.

The diffusion approach has proven to be effective in generating realistic images from prompts or images (Nichol et al., 2021; Rombach et al., 2022; Saharia et al., 2022). DiffusionDet (Chen et al., 2023b) extends the diffusion process by incorporating it into the generation of detection box proposals, while DiffusionDepth (Duan et al., 2025) explores using diffusion models to generate depth images guided by monocular visual conditions. LDM3D (Stan et al., 2023) applies a diffusion model to estimate depth, enabling the simultaneous generation of both an RGB image and its corresponding depth map from a text prompt. Another recent advancement is the use of diffusion to create videos (Ho

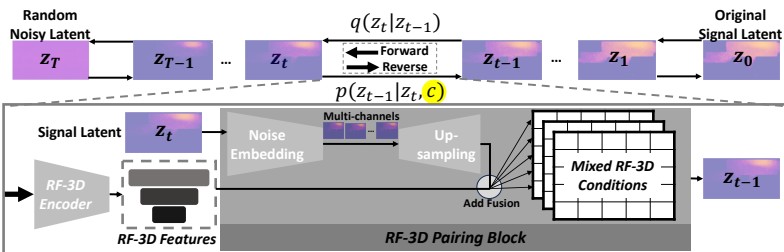

Figure 2: Overview of the diffusion process in **Diffusion**[2]. *RF-3D features* condition the denoising process, while different modalities are fused through the *RF-3D Pairing Block*.

et al., 2022; Singer et al., 2022). Recently, Stable Video Diffusion (Blattmann et al., 2023), Sora (Sora, 2024), and Cosmos (NVIDIA et al., 2025) have demonstrated exceptional performance in this domain.

Most existing research on diffusion in RF signals focuses on generating signal information. RF-Diffusion (Chi et al., 2024) and RF Genesis (Chen & Zhang, 2023) are notable examples of applying diffusion to RF signals. RF-Diffusion uses a diffusion model to generate RF signals across the spatial, temporal, and frequency domains. However, like NeRF[2], RF-Diffusion requires substantial data from the target environment. RF Genesis, on the other hand, combines diffusion models with a ray-tracing approach to generate dynamic 3D scenes and RF signals, with a primary focus on generating data for sensing applications in the mmWave frequency range. In contrast, our diffusion model generates the RF heatmap and supports a broader range of frequencies, including mmWave and Wi-Fi.

## 3 DESIGN OF **Diffusion**[2]

### 3.1 OVERVIEW

**Diffusion**[2] is a generative diffusion model designed to produce realistic RF signal heatmaps from 3D point clouds of indoor environments. The model operates through a forward and reverse diffusion process, where random noise is iteratively transformed into a coherent signal map. This process is guided by a conditioning mechanism using our proposed *RF-3D Encoder* (see Section 3.3), which encodes the geometric and physical context of the environment.

Conditioning is essential for accurately capturing signal propagation effects. Without it, the diffusion model would lack spatial and semantic awareness of the room layout, object locations, or the transmitter (TX) position. Our approach enables the generation of both static heatmaps and temporally consistent heatmap sequences for dynamic scenes.

To this end, we address four key questions: (1) How can physical environments be embedded into a condition vector? (2) How should 2D, 3D, and RF-specific features be represented? (3) How can these signals be fused in the denoising process? (4) How can the system be extended to video-level predictions? We propose: (i) the *RF-3D Encoder* for cross-modal feature extraction, and (ii) the *RF-3D Pairing Block* for integrating them into the diffusion process.

**Condition-guided denoising process.** To incorporate RF signals during the denoising process, we reformulate the reverse function $p(.)$ of diffusion by adding a visual condition $c$ (Duan et al., 2025):

$$p_\theta(z_{t-1}|z_t, \mathbf{c}) := \mathcal{N}(z_{t-1}; \mu_\theta(z_t, t, \mathbf{c}), \Sigma_\theta(z_t, t)) \tag{1}$$

where $z_t$ denotes the noisy signal at timestep $t$, $\mathbf{c}$ is the visual condition representing our *RF-3D Features*. $\theta$ indicates it is trained through neural networks. The design of $c$ is crucial, as it enhances the richness of signals used to capture environmental information, ultimately influencing how accurately the generated RF heatmap reflects real-world scenarios.

### 3.2 *RF-3D Pairing Block*

The *RF-3D Pairing Block* integrates the noisy prediction $z_t \in \mathbb{R}^{H \times W \times C}$ with the environment-aware features generated from the *RF-3D Encoder* to guide denoising. First, the noisy prediction is processed through a noise embedding and upsampling block through noise embedding and upsampling to $\tilde{z}_t = \text{Upsample}(\text{Embed}(z_t))$. This reduces spatial resolution while increasing the number of feature channels, resulting in a compact representation that encodes richer signal semantics suitable for fusion

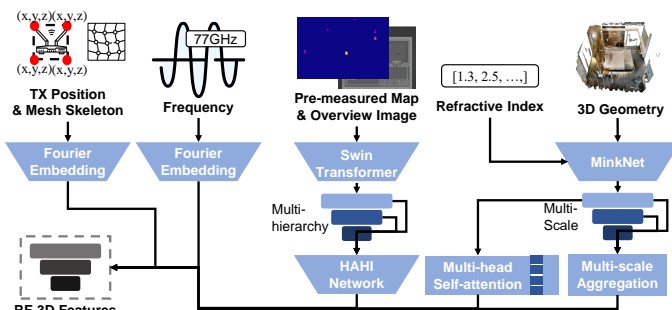

Figure 3: *RF-3D Encoder* embedding 2D, 3D, and RF signal

with the environment-aware features. We then fuse the upsampled latent $\tilde{z}_t$ with the multi-modal features $\mathcal{F}_{\text{RF3D}}$ extracted using the encoder described in Section 3.3 through element-wise addition, $\mathcal{F}_{\text{mixed}} = \tilde{z}_t + \mathcal{F}_{\text{RF3D}}$, where the resulting $\mathcal{F}_{\text{mixed}}$ captures both spatial and temporal characteristics of signal propagation and serves as the input condition $\mathbf{c}$ for Eq. 1. This fusion ensures that the denoising step is informed by both the current prediction state and the surrounding environment. Therefore, this enables the diffusion model to simulate RF signal propagation with geometric consistency and physical plausibility. $\mathcal{F}_{\text{mixed}}$ is then used to compute the next latent state $z_{t-1}$.

### 3.3 *RF-3D Encoder*

The core of our conditioning representation $\mathbf{c}$ is the *RF-3D Features* ($\mathcal{F}_{\text{RF3D}}$), which integrates multi-modal information from 3D geometry, 2D images, and RF signal properties. This encoder extracts semantically and spatially aligned features across modalities to guide the diffusion process.

**3D feature.** Given any point cloud $\mathcal{P} = \{x_i\}_{i=1}^N \subset \mathbb{R}^3$, we use MinkUNet18A (Choy et al., 2019) to extract multi-scale sparse features, denoted as $\mathcal{F}_{3D}^{(l)} = \text{MinkUNet}(\mathcal{P})$ for $l = 1, 2, 3, 4$. To capture hierarchical context and enhance spatial reasoning, we process the multi-level 3D features as:

$$\mathcal{F}_{\text{final}}^{3D} = \text{Interpolate}(\text{MHSA}(\text{FPN}(\{\mathcal{F}_{3D}^{(l)}\}))) \tag{2}$$

where we first apply a feature pyramid network (FPN) (Lin et al., 2017) to merge hierarchical features from different levels, enabling multi-scale contextual understanding. Then, a multi-head self-attention (MHSA) module enhances global spatial reasoning. Finally, since the dimension of 3D features may vary by environment, we interpolate it to obtain a consistent feature size.

We further incorporate material properties by mapping the refractive index to a color embedding at each 3D coordinate, enabling the model to capture signal interactions such as reflection, refraction, and scattering. However, since our multi-scale 3D features from MinkNet encode categorical semantics at each 3D coordinate (*e.g.*, sofa, window), **Diffusion**[2] achieves comparable performance without explicitly relying on refractive index information, which is typically challenging to acquire in real-world environments. This is enabled by the implicit object-level understanding embedded in the 3D feature representation, as discussed in Section 4.4.

**2D feature.** We encode the overview image $I \in \mathbb{R}^{H' \times W' \times 3}$ and the pre-measured heatmap $M \in \mathbb{R}^{H' \times W'}$ using a Swin Transformer (Liu et al., 2021) and fuse their multi-level features via hierarchical aggregation and heterogeneous interaction (HAHI):

$$(\mathcal{F}_I^{2D}, \mathcal{F}_M^{2D}) = (\text{HAHI}(\text{Swin}(I)), \text{HAHI}(\text{Swin}(M))). \tag{3}$$

We then aggregate the two hierarchical representations to obtain the final 2D feature, $\mathcal{F}_{\text{final}}^{2D} = \text{Aggregate}(\mathcal{F}_I^{2D}, \mathcal{F}_M^{2D})$. This modular design mirrors the multi-scale 3D feature processing and enables effective hierarchical reasoning over both visual and RF signal contexts.

**RF signal feature.** We apply Fourier embedding to the transmitter location $\mathbf{b}_{\text{TX}}$, the mesh structure $\mathcal{M}_{\text{mesh}}$ (walls/floors), and the signal frequency $f$:

$$\mathcal{F}_{\text{final}}^{\text{signal}} = \text{Concat}\Big(\phi_{\text{Fourier}}(\mathbf{b}_{\text{TX}}),\ \phi_{\text{Fourier}}(\mathcal{M}_{\text{mesh}}),\ \phi_{\text{Fourier}}(f)\Big) \tag{4}$$

where $\phi_{\text{Fourier}}(x) = [\sin(2^k \pi x), \cos(2^k \pi x)]_{k=0}^{K-1}$. This encoding is well-suited for the sinusoidal nature of RF phase and amplitude modulation, enabling learning across multi-frequency settings.

**Unified condition representation.** We fuse all features to form a complete multi-modal condition, $\mathcal{F}_{\text{RF3D}} = \text{Fuse}(\mathcal{F}_{\text{final}}^{3D}, \mathcal{F}_{\text{final}}^{2D}, \mathcal{F}_{\text{final}}^{\text{signal}})$. The final representation $\mathcal{F}_{\text{RF3D}}$ serves as the conditioning input $\mathbf{c}$ to the denoising distribution in Eq. 1. To fuse it with the upsampled latent $\mathbf{z}_t \in \mathbb{R}^{H \times W \times C}$, we reshape $\mathcal{F}_{\text{RF3D}}$ to match the spatial dimensions $(H, W, C)$.

**Mapping features between 2D and 3D.** *RF-3D Features* integrate information from both 3D point clouds and 2D data, but mapping between these modalities is nontrivial due to inherent ambiguities. Unlike 2D images with fixed resolution, 3D point clouds have a variable number of coordinates and lack structured mappings. Moreover, unlike prior works (Peng et al., 2023; Singh et al., 2023), our setting lacks camera models (e.g., pinhole projection) to directly align 2D and 3D spaces. For 2D-3D alignment, we embed spatial cues into *RF-3D Features*. First, a 2D overview image offers a top-down view with the transmitter (TX) marked as a blue pentagon. Second, we encode the TX bounding box and 3D mesh structures (*e.g.*, walls, floors) using Fourier embeddings to preserve geometric context. These cues guide **Diffusion**[2] in aligning 3D features with the 2D representation.

### 3.4 NETWORK TRAINING

**Transform to signal latent space.** Training diffusion models directly in pixel space is computationally intensive (Rombach et al., 2022). To mitigate this, we follow prior work (Rombach et al., 2022; Duan et al., 2025) and encode the input into a latent signal space before the diffusion process. The model then decodes this latent to generate RF heatmaps. Both encoder and decoder are trained by minimizing signal loss in pixel space, not latent space.

**Loss function.** We train neural networks, denoted as $\theta$ in the reverse process, as shown in Eq. 1. We design our loss with the scaling factor $\lambda$ as follows:

$$L = \lambda_1 L_D + \lambda_2 L_T + \lambda_3 L_{Pre} \tag{5}$$

where $L_D$ denotes the diffusion loss, and $L_T$ captures two pixel-wise losses using both L1 and L2 norms between the prediction and ground truth. $L_{Pre}$ is an RSSI error computed as the mean squared error between the predicted map and the pre-measured input (see Appendix C.1 for details).

### 3.5 GENERATING RF HEATMAP VIDEO

To accommodate dynamic environments, we extend our method to support RF video generation, enabling applications such as network provisioning, diagnosis, and wireless sensing (Zheng et al., 2019; Jiang et al., 2018; Chen & Zhang, 2023). We first extract 3D features from each 3D snapshot. Then we extend the noise model by incorporating a temporal dimension, which needs to be learned concurrently with spatial features. This requires modifications to the U-Net architecture to process temporal correlations. In addition, we introduce temporal layers into the conditioning network to capture the frame-to-frame differences. Specifically, we include multiple frames as input, each containing both *RF-3D Features* and noisy latent $z_t$. Environmental changes, such as variations in human positions, are embedded in the 3D geometry input corresponding to each frame. In addition, we apply a multi-head cross-attention layer that aggregates the 3D features and frame index. This attention mechanism helps identify and highlight relationships between features, such as focusing on the 3D features that dynamically change across frames. For further details, see Appendix C.2.

## 4 EVALUATION

### 4.1 EXPERIMENT SETUP

**RF signal frequency.** We evaluate the model across 10 frequencies in the mmWave band. For Wi-Fi, we include one 2.4 GHz frequency and 10 frequencies in the 5 GHz band. We conduct extensive experiments with various frequency combinations, while maintaining a constant quantity of training data, unless otherwise specified, to ensure fair comparisons. We collect over 55k data samples from diverse 3D environments and frequency ranges, utilizing 80% for training and 20% for testing.

**Comparative methods of amplitude.** We compare amplitudes with the received signal strength indicator (RSSI) values measured at the receivers (RXs). Five baseline schemes are as follows:

- **Ground truth**: We adopt **AUTOMS** (Ma et al., 2024) as the ground truth due to its high accuracy and fast inference, despite its reliance on ray-based computation. As real-world datasets are unavailable, we train our model using this simulated data. Nonetheless, we demonstrate that the trained model achieves comparable accuracy in real-world scenarios.

Figure 4: Wi-Fi signal prediction

Figure 5: mmWave signal prediction

- **NeRF$^2$** (Zhao et al., 2023b): This is the state-of-the-art approach for RSSI prediction, driven by a large training dataset for each 3D environment. In line with the existing NeRF$^2$ setup, we use 5k points as a training dataset to infer the RXs for the entire environment.

- **NeRF$^2$(Fix)**: This variant inherits the structure of NeRF$^2$ but uses the same amount of pre-measurement as ours (*i.e.*, fix the training dataset to contain 15 points and iterate until convergence).

- **MRI** (Shin et al., 2014): An interpolation-based RSSI predictor using a basic propagation model.

- **DiffusionDepth** (Duan et al., 2025): An image diffusion model that generates the depth space from the RGB image. This method incorporates only the 2D features from our *RF-3D Features* to observe the benefits of incorporating 2D, 3D, and RF features.

Although prior works such as RF-3DGS (Zhang et al., 2024), RF-Diffusion (Chi et al., 2024), and RF-Genesis (Chen & Zhang, 2023) also address RF signal generation, a direct comparison with our approach is not feasible due to differing objectives. Specifically, while these methods focus on synthesizing realistic RF signals for a single RX, our model is designed to generate RF heatmaps that capture signal distributions across a large number of RXs.

## 4.2 CHANNEL PREDICTION

### 4.2.1 AMPLITUDE

In both Wi-Fi and mmWave scenarios (Fig. 4, 5), **Diffusion$^2$** outperforms all baselines, achieving median RSSI errors of 1.9 dB (Wi-Fi) and 1.20 dB (mmWave). It reduces errors by 51–72% in Wi-Fi and 54–77% in mmWave across NeRF$^2$, NeRF$^2$(Fix), MRI, and DiffusionDepth. The improvements stem from two key factors: NeRF$^2$ and MRI rely heavily on pre-measured data, causing uncertainty and blurring in unmeasured regions (Fig. 1), and the inclusion of multi-modal data enables **Diffusion$^2$** to better capture the complexities of RF signal propagation. Although DiffusionDepth uses fewer pre-measured points, its limited 2D input restricts its ability to model these propagation effects.

**Importance of multi-frequency dataset.** We find that training on a single frequency limits the model's ability to capture signal–environment relationships. Incorporating data across multiple frequencies, as shown in Fig. 6, allows the model to better understand how signals interact with the environment. Using 10 mmWave frequencies, with just 1/10 of the data per frequency, enables the diffusion process to more accurately mimic signal propagation. This improvement arises from multi-frequency data supporting signal-based diffusion rather than simple image-based diffusion.

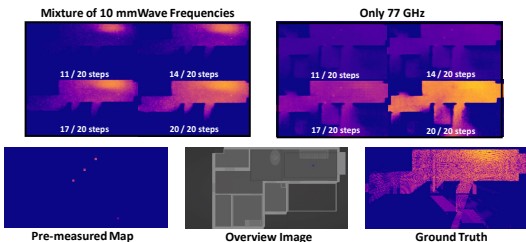

Figure 6: Effect of frequency diversity in training: 10 frequencies (77–77.072 GHz) vs. a single frequency (77 GHz) at 11, 14, 17, 20 diffusion steps

### 4.2.2 AMPLITUDE VIDEO

We compare the amplitude video results with the ground truth at the mmWave frequency. NeRF$^2$, MRI, and Wireless InSite are excluded as they do not support video. **Diffusion$^2$** achieves a median RSSI error of 2.07 dB, effectively capturing dynamic human locomotion and adapting to changes in the 3D environment through video diffusion (see Appendix E.3 for details).

## 4.3 REAL-WORLD SCENARIOS

To validate the practicality of **Diffusion$^2$**, we further examine its performance in real-world environments beyond synthetic data. Specifically, we consider three static indoor scenarios where 3D

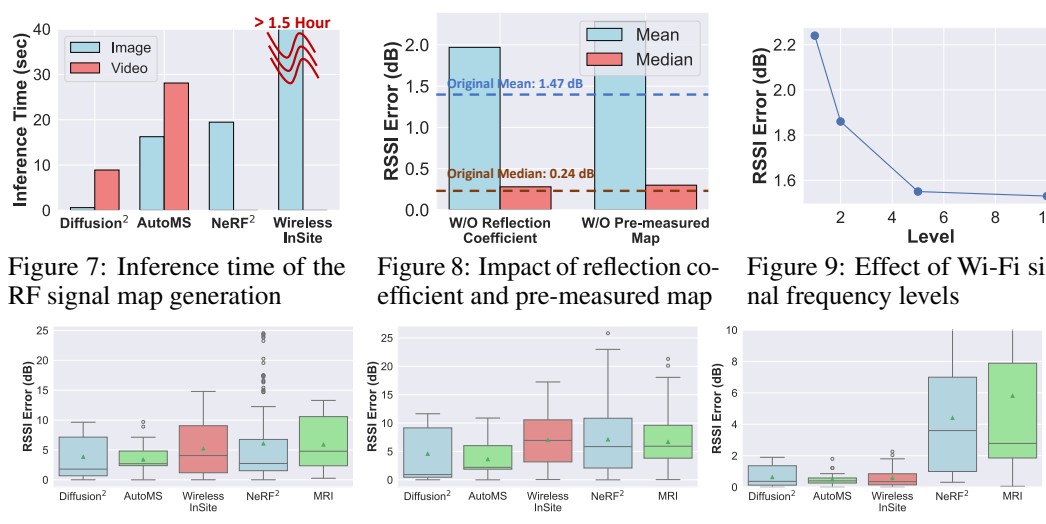

Figure 7: Inference time of the RF signal map generation

Figure 8: Impact of reflection coefficient and pre-measured map

Figure 9: Effect of Wi-Fi signal frequency levels

(a) Scenario I       (b) Scenario II       (c) Scenario III

Figure 10: The RSSI error of real-world scenarios

models are reconstructed using commodity smartphones and RSSI measurements are collected under mmWave frequencies. This evaluation enables us to assess how well **Diffusion**[2] generalizes to realistic deployment conditions and to compare its predictive accuracy against strong baselines.

Fig. 10 summarizes the results, comparing **Diffusion**[2] with AUTOMS, Wireless InSite, NeRF[2], and MRI across the real-world scenarios. Fig. 11 then illustrates the corresponding 3D smartphone scans, measured RSSI, and predicted heatmaps from these methods. **Diffusion**[2] delivers accurate RSSI estimates across diverse locations (*e.g.*, behind walls, outside doors), consistently achieving lower median errors than other methods. Compared to AUTOMS, the strongest baseline, **Diffusion**[2] achieves 0.9 dB, 1.27 dB, and 0.03 dB lower median RSSI across the three scenarios. Against NeRF[2], the improvements are 0.94 dB, 4.93 dB, and 3.23 dB, respectively. We further compare **Diffusion**[2] and AUTOMS on RF video generation, where **Diffusion**[2] achieves comparable accuracy and a slightly better median error of 0.05 dB (see Appendix E.3 for details).

### 4.4 MICRO-BENCHMARKS

**Effectiveness of RF-3D Encoder design.** We conduct ablation testing to assess the impact of each component, as shown in Table 1. The embeddings of 3D features and RF signal features resulted in a performance improvement of approximately 11-23%. Furthermore, the use of multi-scale aggregation and multi-head self-attention on the 3D features led to additional performance improvements of 8-26%. These results demonstrate that each internal component of the *RF-3D Encoder* plays a crucial role in understanding signal propagation.

**Dataset diversity with multiple frequencies.** We explore the significance of incorporating multiple frequencies in the training dataset in Section 4.2. As shown in Fig. 9, we measure the RSSI error by gradually increasing the number of frequencies, referred to as frequency levels, used for training from 2 to 10. Increasing the number of frequencies leads to a 31.69% reduction in RSSI error. We find that using more than 5 frequencies in training is useful for generating accurate RF signal maps.

Table 1: Ablation study of $\mathcal{F}_{\mathrm{RF3D}}$

| Signal Type | Component | RSSI Error (dB) |
|---|---|---|
| Wi-Fi | $\mathcal{F}^{2D}_{\mathrm{final}}$ | 2.63 |
| | $+ \mathcal{F}^{3D}_{\mathrm{final}}$ | 2.32 |
| | $+ \mathcal{F}^{\mathbf{signal}}_{\mathbf{final}}$ | **2.12** |
| mmWave | $\mathcal{F}^{2D}_{\mathrm{final}}$ | 2.43 |
| | $+ \mathcal{F}^{3D}_{\mathrm{final}}$ | 1.85 |
| | $+ \mathcal{F}^{\mathbf{signal}}_{\mathbf{final}}$ | **1.36** |

**Inference without detailed input.** Reflection coefficients and pre-measured maps provide valuable information for estimating RF signal propagation; however, acquiring them in real environments is often challenging. **Diffusion**[2] addresses this limitation by leveraging a pre-trained MinkNet to infer object categories at each 3D coordinate, enabling the diffusion model to produce results comparable to those obtained with full inputs (Fig. 8). When the reflection coefficient is omitted, the mean and median RSSI errors increase by approximately 0.5 dB and 0.04 dB, respectively. Similarly, excluding the pre-measured map raises mean and median errors by roughly 0.81 dB and 0.06 dB. Notably, the

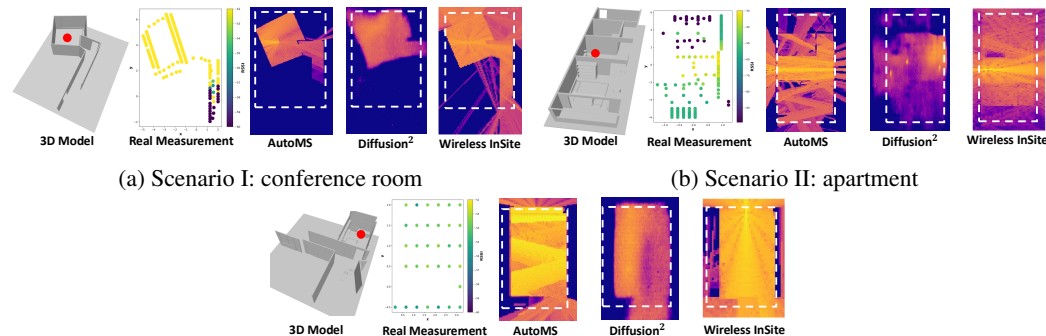

(a) Scenario I: conference room      (b) Scenario II: apartment

(c) Scenario III: office

Figure 11: Overview of real-world scenarios, showing 3D smartphone scans, measured RSSI, and predicted heatmaps from AUTOMS, Wireless InSite, and **Diffusion**$^2$. The AP location is marked by a red circle, and the experiment regions are outlined with white dotted lines.

impact on median error is minimal, and although some localized uncertainty persists without these inputs, the overall quality and fidelity of the generated RF signal maps remain largely intact.

**Robustness against untrained frequencies.** To evaluate frequency generalization across broader bands, we test on an unseen 5.34 GHz signal after training only on 2.4 GHz, 5.16 GHz, and mmWave data. The resulting error of 2.25 dB is comparable to the 2.12 dB error achieved using the full Wi-Fi frequency set. This demonstrates that **Diffusion**$^2$ can effectively generalize to unseen frequencies by leveraging nearby frequency information.

**Inference time.** We measure the average inference time using **Diffusion**$^2$, AUTOMS, NeRF$^2$, and Wireless InSite. As shown in Fig. 7, **Diffusion**$^2$ takes only 0.59 seconds to generate the RF signal image, as its inference time is primarily determined by the neural network size. In contrast, NeRF$^2$ and AUTOMS require about 20 seconds to calculate signals, while Wireless InSite takes over 1.5 hours. In addition, **Diffusion**$^2$ generates an 8-frame video in 8.9 seconds, which is 3.1 times faster than AUTOMS. Importantly, while the computational cost of ray-tracing algorithms like Wireless InSite and AUTOMS increases exponentially with the number of RXs, **Diffusion**$^2$ scales more efficiently. In **Diffusion**$^2$, the number of RXs corresponds to the image resolution, leading to a more gradual increase in inference time as the number of RXs grows.

We further evaluate **Diffusion**$^2$ under three challenging conditions: (i) generalization to operating frequencies unseen during training, (ii) robustness to out-of-distribution material conditions, and (iii) resilience to incomplete 3D inputs from sensing limitations. As detailed in Appendix E.2, **Diffusion**$^2$ maintains low RSSI errors in these scenarios, demonstrating strong generalization to unseen materials, robustness with up to 20% missing 3D points, and reliable performance under frequency shifts.

## 5 LIMITATION

Collecting finely paired 3D and RF datasets across diverse environments is challenging due to dense receiver requirements and labor-intensive setups in each space. As a result, most existing works are limited to small laboratory settings. To overcome this, we use a ray-based simulator to efficiently model complex environments and human motion, enabling faster inference while maintaining realism. While we validate generalizability in three real-world environments, potential distribution gaps between simulated and real data may still impact performance.

## 6 CONCLUSION

We propose **Diffusion**$^2$, an innovative generative diffusion model to estimate RF signal propagation using 3D environments. **Diffusion**$^2$ introduces the novel *RF-3D Encoder* encapsulating the complex 3D point clouds, 2D images, and RF-related features. Then, our *RF-3D Pairing Block* fuses the *RF-3D Features* as the condition to guide the diffusion steps. We further extend our image diffusion to video diffusion to capture temporal changes in the 3D environment. Our extensive evaluations demonstrate the accuracy and efficiency of **Diffusion**$^2$. We incorporate a 3D environment model into the diffusion for the first time to significantly reduce the measurement overhead.

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

## A  ADDITIONAL RELATED WORK

**3D scene understanding.** There has been extensive research in the field of visual 3D scene understanding. Previous studies have primarily focused on training models using accurate 3D labels (Schult et al., 2020; Choy et al., 2019), addressing tasks such as 3D object classification (Wu et al., 2015), 3D object detection (Caesar et al., 2020; Chen et al., 2020), and 3D semantic and instance segmentation (Behley et al., 2019; Huang et al., 2019). OpenScene (Peng et al., 2023) introduces a zero-shot method for understanding 3D scenes with an open vocabulary. This approach leverages CLIP embeddings to calculate dense features for 3D points, co-embedded with text strings and image pixels, to facilitate 3D semantic segmentation.

In comparison to existing ML-based approaches that rely solely on RF measurements, **Diffusion**[2] requires significantly fewer RF measurements for training due to the use of the 3D environment model. 2) By using the 3D environment model as input, it supports various environments without significant measurement overhead or retraining, whereas approaches like NeRF[2] and RF-Diffusion necessitate extensive new measurements and retraining whenever the environment changes. 3) It supports multiple frequencies. 4) It can generate RF heatmaps for both static and dynamic 3D scenes. In short, **Diffusion**[2] combines the strengths of both ray tracing and ML-based approaches to achieve high accuracy, fast performance, flexibility (supporting multiple frequencies and both RF heatmap images and videos), ease of use, and requires minimal training data.

## B  MODELING RF PROPAGATION WITH DIFFUSION MODELS

Predicting radio frequency (RF) propagation is challenging. While the underlying physics is deterministic, real-world environments introduce significant uncertainty from factors like noisy 3D scans, unknown material properties, and complex multipath interference. Consequently, exact, path-based simulations are often computationally intractable, and simple regression models struggle to capture the full range of possible outcomes (Zhao et al., 2023b).

We propose a diffusion-based framework that learns a *distribution over plausible RF fields* rather than predicting a single, deterministic outcome. This approach embraces uncertainty and decomposes the complex problem of field prediction into a sequence of manageable denoising steps. This paradigm has proven effective in other physics-grounded domains, such as world modeling in Cosmos (NVIDIA et al., 2025), by progressively refining an output to ensure it remains physically plausible. Our work extends this concept to RF propagation, demonstrating that diffusion models can accurately fit simulated data while respecting the physical principles of wave propagation.

### B.1  OVERCOMING THE LIMITATIONS OF PRIOR MODELS

Previous attempts to model RF propagation with generative models often fell short. As noted in the NeRF[2] (Zhao et al., 2023b), models like DCGANs and VAEs failed to generalize because they treated RF heatmaps as static spatial *signatures* tied to a transmitter's location. Instead of learning the physics of propagation, they simply memorized geometric patterns. NeRF[2] made progress by incorporating a more physically grounded radiance field representation.

Our model, which we call **Diffusion**[2], builds on this insight. We structure the diffusion process to explicitly mimic the temporal dynamics of wave propagation. As shown in Fig. 6, our model initiates the process with high signal intensity concentrated near the transmitter, which then gradually diffuses outward. This behavior is not just a generative artifact; it is an emergent property that aligns with physical reality.

This physically grounded approach is crucial for learning true propagation semantics. We observed that when key components of our architecture were removed (*e.g.*, in a single-frequency baseline), the model reverted to overfitting, reproducing spatial artifacts of the environment (*e.g.*, apartment layouts) without modeling genuine signal dynamics. In contrast, our full model generates coherent and physically plausible propagation trajectories.

## B.2 THE ADVANTAGE OF A PROBABILISTIC FRAMEWORK

The core advantage of diffusion over deterministic methods is its ability to *represent a distribution over possible RF fields*. This probabilistic approach provides inherent robustness to the uncertainties and incomplete observations common in real-world scenarios, such as material variations or missing geometry in a 3D scan. By learning a range of plausible outcomes, the model generalizes more effectively.

In summary, diffusion offers a physics-aligned and uncertainty-aware framework that bridges the gap between computationally expensive deterministic simulations and brittle pattern-matching approaches.

## C DIFFUSION PROCESS WITH CONDITION

The overall diffusion consists of two processes: **forward** noising $q(.)$ and **reverse** denoising $p(.)$ as shown in Fig. 2.

**Forward process.** Following a Markov chain, a forward process generates $z_t$ starting from the original signal latent $z_0$ by sequentially adding a Gaussian noise distribution $t$ times. The forward process finally generates the random noisy latent $z_T$, which becomes a normal distribution $\mathcal{N}(0, I)$. However, since the diffusion step $T$ is usually set over 1,000, forwarding all steps sequentially is inefficient from a computing resource perspective. So, DDPM applies the reparameterization trick that samples with some steps skipped to process directly from $z_0$ to $z_t$ as follows:

$$q(z_t|z_0) := \mathcal{N}(z_t; \sqrt{\bar{\alpha}_t}z_0, (1 - \bar{\alpha}_t)I) \tag{6}$$

$$:= \sqrt{\bar{\alpha}_t}z_0 + \sqrt{1 - \bar{\alpha}_t}\epsilon \tag{7}$$

where $\alpha_t = \prod_{i=0}^{t} \alpha_i$, $\alpha_t = 1 - \beta_t$, and $\epsilon \sim \mathcal{N}(0, I)$. $\beta$ represents noise variance schedule and $\epsilon$ denotes sampled noise from a normal distribution.

**Reverse process.** In the denoising process, we use the same normal distribution as the forward process and assign the task of predicting a mean $\mu$ and a diagonal covariance matrix $\Sigma$ of the distribution to neural networks as follows:

$$p_\theta(z_{t-1}|z_t) := \mathcal{N}(z_{t-1}; \mu_\theta(z_t, t), \Sigma_\theta(z_t, t)) \tag{8}$$

where $\mu$ denotes the predicted mean of the distribution and $\Sigma$ represents the predicted variance. We append the symbol $\theta$ indicating it is trained through neural networks. With this process, we can finally infer the original signal latent $z_0$ from random noisy latent $z_T$.

**Visual-condition guided denoising process.** To consider RF signal information during the denoising process, we reformulate Eq. 8 by adding a visual condition $c$ (Duan et al., 2025):

$$p_\theta(z_{t-1}|z_t, \mathbf{c}) := \mathcal{N}(z_{t-1}; \mu_\theta(z_t, t, \mathbf{c}), \Sigma_\theta(z_t, t)) \tag{9}$$

The visual condition $c$, which represents our *RF-3D Features*, turns the probability formula in Eq. 8 into a conditional probability Eq. 9. It requires that every step of the diffusion process adheres to the given conditioning $c$, which reflects the real physical environment. The design of conditioning $c$ is critical as it determines whether we can provide rich input signals to feedback environmental information, thereby making the generated RF signal map as consistent with the real scenario as possible. The detailed design of the conditioning $c$ is described in Section 3.3.

**Inference acceleration with DDIM.** DDPM follows a Markov chain, so the inference is slow because generating a single image requires passing $T$, typically over 1,000 diffusion steps. DDIM notices that the objective function of DDPM depends directly on the marginal distribution $q(z_t|z_0)$ not the joint distribution $q(z_{1:T}|z_0)$ and introduces the non-Markov chain to speed up the reverse process with little performance degradation. DDIM reformulates the forward process as follows:

$$q(z_{1:T}|z_0) := q(z_T|z_0) \prod_{t=2}^{T} q(z_{t-1}|z_t, z_0). \tag{10}$$

According to Bayes' theorem, $q(z_{t-1}|z_t, z_0)$ is also a Gaussian distribution, and the mean and variance are determined to ensure $q(z_t|z_0) := \mathcal{N}(z_t; \sqrt{\bar{\alpha}_t}z_0, (1 - \bar{\alpha}_t)I)$ according to Eq. 6 for all $t > 1$ as follows:

$$q(z_{t-1}|z_t, z_0) := \mathcal{N}(z_{t-1}; \tilde{\mu}(z_t, z_0), \tilde{\beta}_t I) \tag{11}$$

where

$$\tilde{\mu}(z_t, z_0) = \frac{\sqrt{\bar{\alpha}_t - 1}\beta_t}{1 - \bar{\alpha}_t}z_0 + \frac{\sqrt{\bar{\alpha}_t}(1 - \bar{\alpha}_{t-1})}{1 - \bar{\alpha}_t}z_t, \tag{12}$$

$$\tilde{\beta}_t = \frac{1 - \bar{\alpha}_{t-1}}{1 - \bar{\alpha}_t}\beta_t. \tag{13}$$

Note that the forward process of DDIM is a non-Markovian process because $z_t$ is dependent on not only $z_{t-1}$ but also $z_0$. We adopt the improved inference process (Song & Ermon, 2020) by fixing the variance schedulers $\alpha$ and $\beta$ during the forward process and setting $\Sigma$ to 0 during the reverse process. In other words, our neural networks focus on predicting $\mu_\theta$ in the denoising process to generate deterministic outputs.

## C.1 NETWORK TRAINING

**Transform to signal latent space.** Training and inference of diffusion models directly based on pixel space require a lot of computing resources and time for optimization (Rombach et al., 2022). Following the latent designs (Rombach et al., 2022; Duan et al., 2025), we encode the pixel space into latent signal space before the diffusion process and decode it backward to generate the RF signal map on a pixel-by-pixel basis. The latent encoder consists of two sequentially connected 2D convolution layers and Tanh as an activation function, while the latent decoder has one sequentially connected 2D transposed convolution layer, one 2D convolution layer, and a sigmoid function as an activation. This transformation into latent space allows for in-depth analysis of the relationships between pixels, which are receivers (RXs) in our problem. The neural networks of the decoder and encoder are indirectly trained by minimizing the signal loss calculated pixel by pixel, not latent space, as shown in Eq. 14.

**Loss function.** We have neural networks to train, denoted as $\theta$, in the reverse process as shown in Eq. 1. Since we set $\Sigma_\theta(z_t, t)$ to 0 for deterministic predictions, we only consider the L2 loss for the denoising prediction and diffusion output, as follows:

$$L_D = ||z_{t-1} - \mu_\theta(z_t, t, \mathbf{c})||^2 \tag{14}$$

where $z_{t-1}$ is calculated based on Eq. 7. We also include two pixel-wise signal losses between the ground truth and the prediction result using L1 and L2 as follows:

$$L_T = \sum_{i,j} |z_0(i,j) - \hat{z}_0(i,j)| + \sum_{i,j} (z_0(i,j) - \hat{z}_0(i,j))^2 \tag{15}$$

where $z_0$ is the ground truth and $\hat{z_0}$ is the predicted signal map. $i$ and $j$ represent the pixel indices. Lastly, we have pre-measured map input that works as the baseline for prediction. So, we apply the mean squared error to calculate point-wise loss between the pre-measured map and our prediction as:

$$L_{Pre} = \frac{1}{N} \sum_{i,j} (p(i,j) - \hat{z}_0(i,j))^2 \tag{16}$$

where $p$ represents the pre-measured signal map and $N$ is the number of actually measured points in $p$. Finally, we get our loss with the scaling factor $\lambda$ as follows:

$$Loss = \lambda_1 L_D + \lambda_2 L_T + \lambda_3 L_{Pre}. \tag{17}$$

(a) Two temporal layers denoted as T1 and T2 in processing both *RF-3D Features* and noisy latent $z_t$

(b) A multi-head cross-attention layer for understanding the relationship between frames

Figure 12: Diffusion model for video generation

## C.2 Generating RF heatmap video

Following the approach in (Rombach et al., 2022), we incorporate one Conv3D temporal layer and one temporal attention layer for both the noisy images $z_t$ and the *RF-3D Features*, labeled T1 and T2 in Fig. 12a. While the spatial layer processes information within each individual frame, these two temporal layers manipulate the feature dimensions across frames. Specifically, the initial input shape is $(b, f, c, h, w)$, where $b$ is the batch size, $f$ is the frame index, $c$ is the image channel, and $h$ and $w$ are the height and width of the images. The spatial layer processes this input for each frame individually.

The Conv3D temporal layer reshapes the input in the following steps:

$$(b, f, c, h, w) \rightarrow (b, \mathbf{c}, \mathbf{f}, h, w) \rightarrow (b, f, c, h, w)$$

The temporal attention layer reshapes as follows:

$$(b, f, c, h, w) \rightarrow (b, \mathbf{h}, \mathbf{w}, \mathbf{f}, \mathbf{c}) \rightarrow (b, f, c, h, w)$$

These temporal layers mix the features across frames by reordering the dimensions, rather than stacking layers across frames or batch units. Importantly, although these layers perform reshaping internally, the final output shape matches the original input shape, allowing these temporal layers to be seamlessly integrated into the architecture without altering the existing design.

## D Implementation details

**3D dataset.** Our problem requires a 3D environment dataset that can be used to place the transmitter (TX) in the appropriate position. Therefore, each object should be stored separately to facilitate manipulation. However, popular datasets like Matterport (Sulaiman et al., 2020) and ScanNet (Dai et al., 2017) only offer a unified mesh file for the entire environment, lacking the desired granularity. Consequently, we adopt 3D-FRONT (Fu et al., 2021), a dataset that aligns with our requirements and features synthetic indoor scenes with professionally crafted layouts, encompassing 18,797 rooms with diverse objects.

**3D dataset augmentation.** 3D-FRONT provides about 18K rooms, but the structure of each room is quite similar, and the number of datasets is not enough, limiting its ability to train our large-scale diffusion model. Therefore, we apply two data augmentation methods. First, the structure of the 3D-FRONT dataset contains one apartment, which is divided into several rooms such as a living room and bathroom. We extract different rooms from the apartments and generate more apartments by randomly combining rooms. Second, our environment requires one TX to be located. Therefore, we enhance the diversity of the dataset by randomly placing TXs inside the room. In particular, this augmentation is suitable for our problem because signal propagation plays a crucial role in predictions both indoors and outdoors. As a result, we secure over 55k rooms with a variety of layouts and an appropriately located TX.

**RF signal dataset.** We utilize the wireless channel simulator of AutoMS (Ma et al., 2024) to generate the amplitude and phase of the 3D environments. We generate the RF signal map for both Wi-Fi and mmWave considering the multiple channels. For Wi-Fi, we consider 2.4 GHz and 10 different channels for 5 GHz as follows: 5.16, 5.18, 5.20, ..., and 5.34 GHz. For mmWave, we divide into 10 different channels based on the frequency equation within each sweep (Mao et al., 2016): $f = f_{min} + \frac{B \times t}{T_c}$ where $B$ is the signal bandwidth, $t$ is a sweep index, and $T_c$ is the chirp length. $t$ is determined by sampling rate $R_s$ as $[0 : 1/R_s : T]$. All variables except $T_c$ are fixed according to the

board specifications, *i.e.*, $B = 4e9$, $S_r = 25e6$, and $f_{min} = 77e9$. We set $T_c$ as $20e\text{-}6$ to chirp into 501 frequencies from mmWave and select the first 10 frequencies for our dataset, *i.e.*, 77, 77.008, 77.016, ..., and 77.072 GHz. We extensively evaluate **Diffusion**[2] by varying the combinations of frequencies in the input dataset, such as using 1 to 10 frequencies. Note that we fix the total number of training datasets across all evaluations to ensure a fair comparison.

**2D feature.** The Swin Transformer (Liu et al., 2021) is employed to generate visual conditions as multi-scale layers for the overview image and pre-measured map, and we incorporate these features using a hierarchical aggregation and heterogeneous interaction (Li et al., 2023). This multi-scale feature embedding is particularly effective for RF signal estimation because it spans small to large scales, similar to signal propagation properties. Additionally, a feature pyramid neck (FPN) (Lin et al., 2017) is utilized to consolidate features into diffusion conditions. For the Swin Transformer, we specify channel dimensions as [192, 384, 768, 1536]. Also, we randomly choose 15 points in the RF signal map for the pre-measured map.

**3D feature.** We use the pre-trained MinkNet model (Choy et al., 2019) with 21 classes for 3D geometry embedding. We use four levels of multi-scale features before the final layer and apply the FPN for these features to align with the 2D multi-scale embedding. We then use interpolation to unify the feature size at each level. The interpolation size is $(fea, coords) = (64, 30000)$, where $fea$ is the feature size and $coords$ represents the 3D coordinates. In addition, we apply multi-head self-attention for the last layer from the MinkNet model using Perceiver IO (Jaegle et al., 2021). We use 512 latent dimensions and 12 heads for cross-attention and latent self-attention.

**Hyperparameters.** We employ the PyTorch framework (Paszke et al., 2019) and conduct training with a batch size of 16 over 20 epochs with a single NVIDIA A100 GPU. We use the Adam optimizer (Kingma & Ba, 2014) and a linear learning rate warm-up strategy for the first 15% of iterations. The initial learning rate is $10^{-3}$ and decreases sequentially over 10, 15, and 20 epochs, applying a multiplicative gamma factor of 0.8, 0.2, and 0.04, respectively. We set an equal ratio of $L_D$, $L_T$, and $L_{\text{Pre}}$ in the loss function.

**Diffusion setup.** We use the improved sampling process (Song & Ermon, 2020) with 1,000 diffusion steps for training and 20 inference steps for inference. The learning rate is $10^{-4}$ for image diffusion and $10^{-3}$ for video diffusion. The maximum signal strength of the amplitude is 70 for all experiments. The resolution of the results is $352 \times 705$ and $52 \times 72$ for image and video, respectively. Our video generation model outputs 8 frames. Our model requires approximately 40 GB of GPU memory during training and completes training in about one day.

**Human locomotion dataset.** To collect a dataset for video diffusion, we use DIMOS (Zhao et al., 2023a), which generates human locomotion in a 3D environment. DIMOS uses a Markov decision process to create reasonable human movements while avoiding collisions between surrounding objects. We extract 8 snapshots for each room through DIMOS and generate an amplitude map according to each snapshot environment through the wireless channel simulator (Ma et al., 2024).

# E  EVALUATION DETAILS

## E.1  REAL-WORLD MEASUREMENT SETUP

We conduct experiments in three indoor scenarios as shown in Fig. 11. We use Polycam (Polycam, 2025) to obtain the 3D models of the experiment environment. We use two Acer Travelmate P658 laptops with Qualcomm QCA6320 chipset-based 60 GHz commercial Wi-Fi cards to measure the mmWave received signal strength indicator (RSSI). The access point (AP) and station use a $6 \times 6$ uniform planar array (UPA) with a $120°$ field-of-view (Song et al., 2023) and 4 corner antennas deactivated. The antenna element spacing is $0.58\lambda$ (Zhao et al., 2020). Each antenna has a 1-bit switch (on or off) and a 2-bit phase shifter. All antennas share a single RF chain. The central carrier frequency is 60.48 GHz. We also conduct real measurements for the RF signal video where an object 1.5 meters in height moves in Scenario III. We place the wireless receiver at designated locations for measurement.

Table 2: Robustness to unseen frequencies

| Trained Frequencies (GHz) | Test Frequency (GHz) | RSSI Error (dB) |
|---|---|---|
| 77-77.024, 77.040-77.072 | 77.032 | 1.52 |
| 2.4, 5.16, 77-77.072 | 5.34 | 2.25 |

Table 3: Generalization to unseen materials

| Material Replacement Ratio | RSSI Error (dB) |
|---|---|
| 0% (baseline) | 1.36 |
| 10% | 1.40 |
| 30% | 1.53 |
| 50% | 1.68 |

Table 4: Robustness to incomplete 3D data

| 3D Input Removed Ratio | RSSI Error (dB) |
|---|---|
| 0% (baseline) | 1.36 |
| 5% | 1.37 |
| 10% | 1.40 |
| 20% | 1.48 |
| 40% | 1.87 |

### E.2 MICRO-BENCHMARKS

**Robustness against untrained frequencies.** We evaluate the ability of **Diffusion**[2] to infer RF signal maps for frequencies not included in the training set, as presented in Table 2. When excluding the 77.032 GHz frequency from a set of 10 mmWave frequencies, the mean RSSI error is 1.52 dB, which is comparable to the 1.36 dB error when the full mmWave set is used. Furthermore, to assess frequency generalizability across wider bands, we test on an unseen 5.34 GHz frequency spanning both Wi-Fi and mmWave ranges. The resulting error is 2.25 dB, closely aligned with the 2.12 dB error observed when using the complete Wi-Fi frequency set. These results indicate that **Diffusion**[2] can effectively generalize to unseen frequencies by leveraging information from adjacent frequency datasets.

**Generalization across unseen material conditions.** In real-world scenarios, the electromagnetic characteristics of objects vary substantially with their material composition, leading models to inevitably face unseen material distributions at deployment. To rigorously assess this generalization capability, we constructed a dedicated test set in which object materials differ from those in the training set (*e.g.*, walls replaced with plasterboard instead of concrete/brick). Without any fine-tuning, the pretrained model exhibits only a gradual increase in RSSI error with higher material replacement ratios, remaining below 1.7 dB, thereby demonstrating strong robustness to out-of-distribution material conditions.

**Robustness to incomplete 3D data.** In real-world deployments, 3D input data are often incomplete due to sensing limitations and occlusions. To evaluate robustness under such conditions, we randomly removed 3D input points for the FMCW signal. The model remains robust up to 20% missing data, exhibiting only a modest increase in error. This robustness can be attributed to MinkowskiNet's sparse convolutional architecture, which effectively handles incomplete and irregular 3D inputs.

### E.3 AMPLITUDE VIDEO

We compare amplitude video results with the ground truth using the mmWave signal frequency, as shown in Fig. 13. The real-world measurement in Scenario III is shown in Fig. 13a, while the simulated mmWave result is presented in Fig. 13b. NeRF[2], MRI, and Wireless InSite are excluded, as they do not support video output. In the real-world evaluation, **Diffusion**[2] achieves comparable accuracy and a slightly improved median error, outperforming AUTOMS by 0.05 dB. On the simulated dataset, **Diffusion**[2] attains a median RSSI error of 2.07 dB, effectively captures dynamic human locomotion, and adapts flexibly to changes in the 3D environment through our video diffusion.

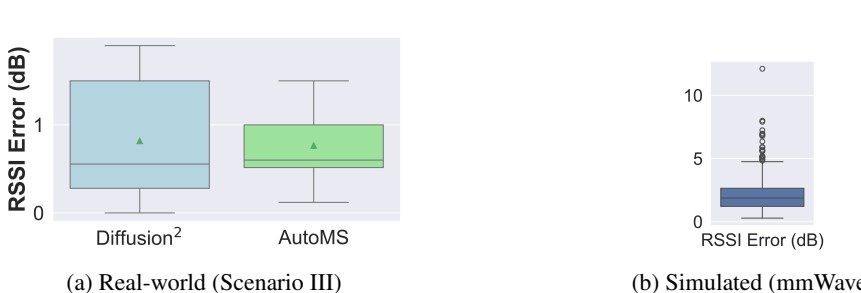

(a) Real-world (Scenario III)  (b) Simulated (mmWave)

Figure 13: Evaluation of amplitude video generation for simulated and real-world measurements.

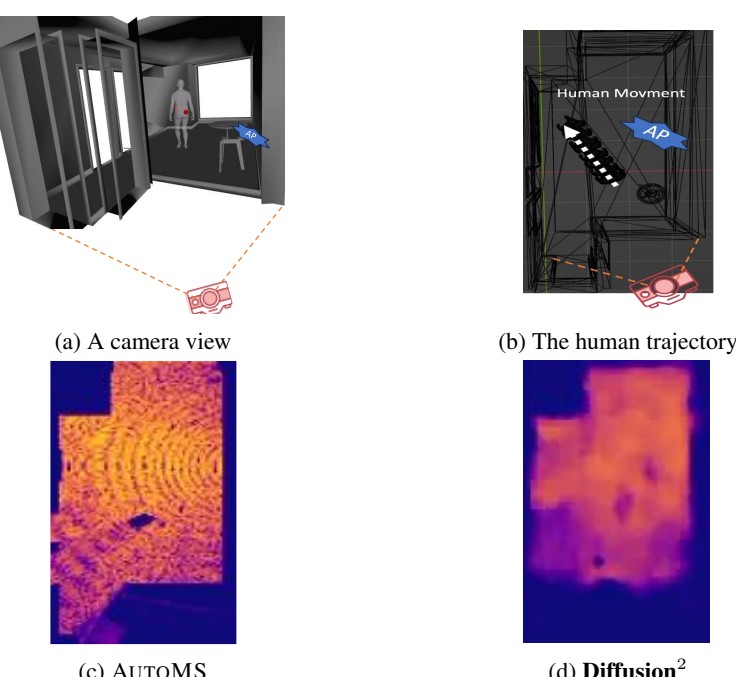

(a) A camera view  (b) The human trajectory

(c) AUTOMS  (d) **Diffusion**$^2$

Figure 14: Video diffusion examples from synthetic dataset. (c) and (d) are snapshots from the video.

