# OpenReview forum: "Diffusion^2: Turning 3D Environments into Radio Freqneucny Heatmaps"
_ICLR.cc/2026/Conference — ICLR 2026 Conference Withdrawn Submission_

### Official Review · Reviewer_QhzM · 2025-10-29

**Soundness:** 3
**Presentation:** 2
**Contribution:** 2
**Rating:** 4
**Confidence:** 4

**Summary:**

Diffusion2 is a generative diffusion model that predicts high-fidelity RF heatmaps from 3D point clouds with only 15 pre-measurements, supporting Wi-Fi and mmWave bands. It introduces an RF-3D Encoder for multi-modal fusion and an RF-3D Pairing Block for conditioned denoising, enabling accurate multipath modeling and dynamic scene generation.

**Strengths:**

+ The diffusion-based methodology provides a solid foundation for future multimodal unification. Its probabilistic denoising framework supports flexible conditioning on diverse inputs such as 3D geometry, 2D images, and RF signals, enabling seamless integration of additional modalities like audio, inertial, or environmental data to enhance holistic wireless scene understanding in 6G applications.

**Weaknesses:**

- While Diffusion2 applies diffusion models to RF heatmap generation from sparse 3D points, its novelty is modest. The method mainly reuses standard DDPM/DDIM pipelines, latent diffusion encoding, and simple additive conditioning without introducing new RF-specific mechanisms. Its modules, such as the RF-3D Pairing Block and RF-3D Encoder, assemble existing components like MinkUNet, Swin Transformer, and Fourier embeddings, mirroring prior multi-modal diffusion works.


- While the RF-3D Encoder presents an innovative fusion of multi-modal inputs, including 2D images, 3D geometry via MinkUNet, and RF signals through Fourier embeddings, its use of simple element-wise addition for feature fusion may fail to capture complex non-linear relationships among modalities. This simplification can cause misalignment or feature dilution in cluttered or heterogeneous environments. Although ablation results show that each modality improves accuracy, the lack of cross-modal attention or comparisons with more advanced fusion mechanisms leaves uncertainty about the model’s robustness and generalization in real-world scenarios such as lighting variations or occlusions.

- The paper omits an essential baseline by not comparing with WRF-GS (Wen et al., 2024), which shares the same objective as Diffusion2 and NeRF2. If Diffusion2 can be evaluated against NeRF2, it should likewise be compared with WRF-GS, which performs comparable tasks at a significantly lower runtime. Without this comparison, it is difficult to assess whether Diffusion2 achieves comparable or superior efficiency and accuracy.



- Its practicality is constrained by low output resolution and restrictive input requirements. The generated heatmaps with resolutions of 352×705 for images and 52×72 for videos are too coarse for dense receiver grids or detailed multipath analysis, which limits their value for high-precision 6G diagnostics. Data collection using smartphone-based LiDAR scans introduces variability caused by occlusions and lighting conditions. The fixed use of 15 pre-measured points also appears arbitrary, as performance is sensitive to this choice without adequate justification or analysis.

**Questions:**

Please see the points raised in the Weaknesses section.

---

### Official Review · Reviewer_FByQ · 2025-10-31

**Soundness:** 3
**Presentation:** 3
**Contribution:** 3
**Rating:** 4
**Confidence:** 3

**Summary:**

This paper proposes Diffusion^2, a diffusion-based model that generates RF heatmaps from 3D point clouds. The key contribution is the RF-3D Encoder, which effectively integrates 3D geometry, 2D overview images, and RF signal features to condition the diffusion process. The method achieves high accuracy (1.9 dB error) and significant speedup (27×) compared to existing approaches, while requiring only minimal pre-measurements (15 points). The extension to video generation for dynamic environments further enhances its practical value.

**Strengths:**

First to use diffusion models for RF heatmap generation from 3D point clouds. Strong empirical results with minimal measurements and fast inference.

Bridges generative AI and wireless communication, enabling efficient and scalable RF modeling.

The paper is well-organized and easy to follow.

**Weaknesses:**

The use of a ray-tracing simulator (AutoMS) as ground truth may limit the validity of “accuracy” claims in real physical environments.

The sensitivity of performance to the location of the 15 pre-measurement points is not thoroughly analyzed.

**Questions:**

1. How sensitive is the model’s performance to the spatial distribution of the 15 pre-measurement points? Would poorly distributed points harm the results?
﻿
2. Since the ground truth is generated by a simulator, to what extent does the model capture true physics versus approximating the simulator’s behavior?
﻿
3. Does the model’s performance depend heavily on the pre-trained 3D segmentation model (MinkNet)? How does it generalize to unseen object categories?

---

### Official Review · Reviewer_m8tc · 2025-11-01

**Soundness:** 3
**Presentation:** 2
**Contribution:** 3
**Rating:** 6
**Confidence:** 3

**Summary:**

- The paper proposes a novel approach to conditionally generate RF heatmaps (RSSI specifically) based on 3D point cloud information.
- The crux of the proposed is leveraging a 2-stage diffusion model (akin to Stable Diffusion and DiffuionDepth).
- A specific novelty is use of 3D scene-specific data to provide conditional guidance during generation using a "RD-3D Encoder" that constructs a 3D-based feature map (using some 3D knowledge of the scene e.g., mesh reconstruction) and injects this at each step of the denoising process.
- The approach is evaluated both on synthetic (AutoMS) and real-world measurement data.
- Evaluation on a range of frequencies, static and dynamic scenes, synthetic and real data shows promising improvements (esp. in data-limited regimes) when compared to relevant baselines (e.g., NeRF^2)

**Strengths:**

1. The "hybrid" nature of the approach is well-motivated and practical i.e., using both 3D scene priors (much like conventional ray tracers) and data-driven learning (in this case with diffusion models)
2. The manner in which 3D priors are leveraged is compelling -- the colors and visual features are used (which provide a good proxy to underlying RF material properties). Ablations also indicate these provide meaningful gains.
3. The evaluation is comprehensive and addresses a range of scenarios (e.g., multi-frequency, dynamic scenes, limited data)

**Weaknesses:**

**1. RF signal features**
- It appears that only the location of the tx/rx is considered (L265-269). However, RF propagation is highly-directional (esp. in mmWave scenarios). It is unclear how the approach handles or generalizes to these nuances (esp. when tx and rx antenna boresight direction is not aligned)
- Although this term is supposed to capture RF features, it also appears to include mesh. It is not justified why duplicating (or perhaps even providing richer information compared to 3D point clouds) is necessary. Importantly, this term appears quite critical and might overshadow the 2D/3D features. It would be interesting to re-run the ablation (Table 1) in reverse order i.e., evaluating marginal gains of 2D/3D features over the RF signal term (which already contains 3D information).

**2. Training and Evaluation data - details unclear**
- The training and evaluation data details is largely unclear, and this is an obstacle to better understanding the evaluation setting.
- Specifically, L315-317 mentions "55k samples from diverse 3D environments". Many details are unclear e.g., whether dataset is publicly available, are they indoor/outdoor, number of samples per environment.
- Given that the data underpins the paper's good empirical results, I believe it is crucial to elaborate on the data.
- Furthermore, L321-323 also mentions that the ground-truth data is derived from "AutoMS" approach. This appears to imply that the proposed approach is evaluated on predictions of another model, and the accuracy is highly dependent on how accurate AutoMS is.

**3. Relies on high-quality training data and generalization**
- L317 remarks split of the data: 80-20% train-test split. It is unclear how the split is done (is it IID?)
- Importantly, is part of the model evaluation performed in scenes and measurements that appeared during training? In which case, this would undermine the claims around using "15 points".

**Questions:**

1. Does the approach account for directional antenna?
2. Why does the RF signal feature term contain the mesh?
3. Please elaborate on the details of training and test data.
4. "15 training datapoints": how are these sampled? How is this different from the 80% of the 55k sample dataset?

---

### Official Review · Reviewer_euLv · 2025-11-01

**Soundness:** 3
**Presentation:** 3
**Contribution:** 3
**Rating:** 0
**Confidence:** 3

**Summary:**

[Will fill in 12h if SAC/PC reviews Non-Anonymous issue and decides to move forward with reviewing.]

**Strengths:**

[Will fill in 12h if SAC/PC reviews Non-Anonymous issue and decides to move forward with reviewing.]

**Weaknesses:**

[Will fill in 12h if SAC/PC reviews Non-Anonymous issue and decides to move forward with reviewing.]

**Questions:**

[Will fill in 12h if SAC/PC reviews Non-Anonymous issue and decides to move forward with reviewing.]

---

### Note · Authors · 2025-11-14

I have read and agree with the venue's withdrawal policy on behalf of myself and my co-authors.